# Whole-Genome Sequencing of *Mycobacterium tuberculosis* Isolates from Diabetic and Non-Diabetic Patients with Pulmonary Tuberculosis

**DOI:** 10.3390/microorganisms11081881

**Published:** 2023-07-26

**Authors:** Ranjitha Shankaregowda, Yuan Hu Allegretti, Mahadevaiah Neelambike Sumana, Morubagal Raghavendra Rao, Eva Raphael, Padukudru Anand Mahesh, Lee W. Riley

**Affiliations:** 1School of Public Health, Division of Infectious Diseases and Vaccinology, University of California, Berkeley, CA 94720, USA; ranjitha@jssuni.edu.in (R.S.); lwriley@berkeley.edu (L.W.R.); 2Department of Microbiology, JSS Medical College and Hospital, JSS AHER, Mysore 570015, India; mnsumana@jssuni.edu.in (M.N.S.); morubagalrao@jssuni.edu.in (M.R.R.); 3School of Public Health, Division of Epidemiology, University of California, Berkeley, CA 94720, USA; yuan_hu@berkeley.edu; 4Division of Epidemiology and Biostatistics, School of Medicine, University of California, San Fransico, CA 94143, USA; eva.raphael@ucsf.edu; 5Department of Respiratory Medicine, JSS Medical College and Hospital, JSS AHER, Mysore 570015, India

**Keywords:** *Mycobacterium tuberculosis*, whole-genome sequencing, phylogenetic analysis

## Abstract

The association of tuberculosis and type 2 diabetes mellitus has been a recognized re-emerging challenge in management of the convergence of the two epidemics. Though much of the literature has studied this association, there is less knowledge in the field of genetic diversities that might occur in strains infecting tuberculosis patients with and without diabetes. Our study focused on determining the extent of diversity of genotypes of *Mycobacterium tuberculosis* in both these categories of patients. We subjected 55 *M. tuberculosis* isolates from patients diagnosed with pulmonary TB with and without type 2 diabetes mellitus to whole-genome sequencing on Illumina Hi Seq platform. The most common lineage identified was lineage 1, the Indo-Oceanic lineage *(n* = 22%), followed by lineage 4, the Euro-American lineage (*n* = 18, 33%); lineage 3, the East-African Indian lineage (*n* = 13, 24%); and lineage 2, the East-Asian lineage (*n* = 1, 2%). There were no significant differences in the distribution of lineages in both diabetics and non-diabetics in the South Indian population, and further studies involving computational analysis and comparative transcriptomics are needed to provide deeper insights.

## 1. Introduction

Globally, about 10 million people are estimated to have developed tuberculosis (TB) in 2020 with eight countries accounting for two-thirds of the global total, with India reporting the largest proportion at 26% [1]. Type 2 diabetes mellitus (DM) is becoming a major public health problem globally, especially in emerging-economy countries [2]. Worldwide, about 15–25% of annual incident TB cases are estimated to occur among persons with DM [3,4]. World Health Organization (WHO) has indicated that India will become the “diabetes capital of the world” by 2025 [5]. DM increases the risk of developing TB by two- to three-fold [6,7], and DM worsens the clinical course of TB, while TB worsens glycemic control in those with DM [5]. Also, DM is associated with treatment failure, relapse, and deaths [8]. The association of active TB in DM patients is significantly increased when compared to those without DM [9,10]. Diminished innate and adaptive immunity likely contributes to the reduced ability of DM patients to control *M. tuberculosis* infection [11]. In DM patients, subsequent episodes of infection are more likely to be from the same bacteria as the previous episode. However, previous studies have shown that the occurrence of exogenous re-infection by another strain is observed in one-fifth of the cases [12]. Also, the idea that genetically diverse strains display distinct transmission dynamics even within the same community could support the hypothesis that there is a propensity for TB-DM patients to be infected by a specific lineage. A study conducted in North Lima, Peru, found that dysglycemia could predispose to a specific lineage. Specifically, the authors found that Beijing and Haarlem strains were more common among diabetics than non-diabetics [13]. Nevertheless, to confirm their findings, they suggested further studies, aimed to study the transmission of specific lineages in the TB population with and without DM, and concluded that it is possible that different strains have different transmission dynamics and that some strains may be more easily transmitted among DM patients when compared to non-DM patients. Therefore, there is a need for similar studies in different countries across the world to confirm whether such findings of different lineages among subjects with and without diabetes are generalizable to other countries. Here, we analyzed whole-genome sequences (WGS) of *M. tuberculosis* isolates from newly diagnosed TB patients in a high-burden city in India to determine if there was any geographic clustering by DM status, and also to assess if those with DM were more likely to be infected with a greater diversity of *M. tuberculosis* lineages than those without DM.

## 2. Materials and Methods

### 2.1. Study Design

Patients diagnosed with pulmonary TB examined between January and December 2019 at J.S.S. Hospital in Mysore, Karnataka state, India, were included in the study. We grouped the patients according to their place of residence in different administrative units of the state, i.e., districts and taluks including Mysore, Chamrajnagar, K.R. Pete, Hassan, Hunsur, Kodagu, Mandya, Srirangapatna, T. Narasipura, and H.D.Kote. Sputum samples from the recruited subjects were tested at the clinical Microbiology laboratory at J.S.S Hospital. Samples that were smear-positive by Ziehl Neelsen staining were utilized for the study. Patient details were collected from the hospital records.

### 2.2. Sample Processing

Sputum samples were digested and decontaminated with the N-acetyl-L-cysteine sodium hydroxide (NALC-NaOH) method and cultured on Lowenstein–Jensen medium (HiMedia Laboratories, Mumbai, India). Cultures were monitored for growth for up to 4 weeks before declaring them as negative growth. *M. tuberculosis* colonies were scraped from the Lowenstein–Jensen medium and genomic DNA was extracted with the QIAamp DNA mini kit (QIAGEN, Hilden, Germany) following the manufacturer’s protocol. Relevant patient demographic and clinical information details were collected from their medical records.

### 2.3. DNA Quality Check

The DNA samples were assessed for quality with the criteria of an OD260/280 ratio of 1.8 to 2.0 and an OD260/230 ratio of 2.0 to 2.2, with the latter bearing an additional value for purity. In addition, gel electrophoresis was performed to determine if a further cleaning was required before shearing.

### 2.4. DNA Library Preparation Protocol

Whole-genome libraries were prepared with NEBNext^®^ Ultra™ DNA Library Prep Kit (Cat. No: E7370L). The workflow involved shearing of DNA (250 bp), repairing ends, and adenylation of 3′ ends, followed by adapter ligation. At each step, the products were purified with AMPure beads (Beckman Coulter, Cat. No.: A63882, San Jose, CA, USA). The adapter sequences were added onto the ends of DNA fragments to generate paired-end libraries. The resulting adaptor-ligated libraries were purified and index tags were added by amplification, followed by purification. Libraries were assessed for quality and quantity with an Agilent 2200 tapestation system (Cat. No.: 3-PM 863NA).

### 2.5. Sequencing Protocol

Prepared libraries were quantified with Qubit high-sensitivity reagent. The obtained libraries were diluted to a final concentration of 2 nm in 10 μL and were subjected to cluster amplification. Once the cluster generation was completed, the flow cells were loaded on to the sequencer. The sequencing was carried out on Illumina Hi Seq at MedGenome Labs Ltd., Bangalore, to generate 2 × 150 base-pair sequence reads at 100× sequencing depth coverage (~0.5 GB). Sequence data were processed to generate FASTQ files, which were uploaded onto the FTP server for secondary data analysis.

### 2.6. Bioinformatics Analysis

The FASTQ sequence files were analyzed with Geneious [14] software version R11.02 with default settings. Raw and processed read statistics were computed and visualized with FastQC v.0.11.3 for quality control. Reads with Phred score over 30 were used for our analysis. The short reads were assembled by mapping to a reference sequence of H37Rv *M. tuberculosis* strain (Gen Bank accession number AL_123456) and de novo assembly. Paired-end reads were mapped to the publicly available annotated genome of *M. tuberculosis* reference strain H37Rv (Gen Bank accession number AL_123456). The consensus sequence was obtained with the default setting of Geneious. Consensus fasta files were submitted for rapid annotation based on the subsystems technology (RAST) [15] server and genotyping software. Mycobacterial lineage, sublineage, main spoligotype, and RD (region of difference) were predicted with TB Profiler (https://tbdr.lshtm.ac.uk/, accessed on 17 August 2019) with the default setting. Lineage prediction, Beijing typing, and in silico spoligotyping were performed with CASTB at default settings [16], and TB Profiler and PhyresSE [17] databases were consulted for the detection of drug resistance and associated mutations. Core and accessory genome phylogenetic trees were constructed with the automasublineated pipeline of Roary [18] with the default setting. The newick files of the trees were visualized with the iTOL v5 (Interactive tree of life) online-based software (https://itol.embl.de/, accessed on 25 February 2020) [19].

### 2.7. Statistical Analysis

Data were entered in Microsoft Excel and analyzed using SPSS Version 25 (licensed to the institution). Descriptive analysis was carried out using proportions, and graphs were used to describe the lineage distribution. Inferential statistics were analyzed using chi-square analysis to find the association between patient characteristics and diabetes status. *p*-values were considered significant when <0.05.

## 3. Results

### 3.1. Patient Characteristics

Between January and December 2019, we obtained 55 *M. tuberculosis* isolates from patients diagnosed with pulmonary TB. Of these, 20 (36%) were from patients with DM and 35 (64%) were from patients without DM. Demographic and clinical characteristics of the patients are shown in Table 1. *p* value was calculated via chi-square test with a 2 × 2 contingency table and the association between two categorical variables was assessed. Among all the variables tested, only the *p* value for age group was found to be significant (0.04). The majority of patients were men (*n* = 40, 73%) in the age group of 50–69, with diabetes being the major co-morbidity (*n* = 20, 36.36%). No formal education of any grade had been received by most patients (*n* = 24, 43.63%) and many of them were unemployed or their occupation status was unknown (*n* = 21, 38%). There was a history of smoking in only a small number of patients *(n* = 7, 12.7%). 

### 3.2. M. tuberculosis Genotypes

To explore the genomic diversities of the isolates, we performed whole-genome sequencing to characterize the genomic profiles of the 55 isolates. Figure 1 shows the distribution of lineages. The most common lineage among the samples was lineage 1, the Indo-Oceanic lineage (*n* = 22, 40%), followed by lineage 4, the Euro-American lineage (*n* = 18, 33%); lineage 3, the East-African Indian lineage (*n* = 13, 24%); and lineage 2, the East-Asian lineage (*n* = 1, 2%). One isolate was not assigned any lineage by the bioinformatics tools as it was identified as contaminated. Indo-Oceanic lineage was the most common lineage among both DM (*n* = 9, 45%) and non-DM patients (*n* = 13, 37%).

Table 2 and Figure 2 show distribution of the study population with regard to lineages and their geographical place of residence. Each district had almost equal number of diabetic and non-diabetic patients. The highest number of patients came from Mysore (*n* = 26). All four lineages were detected in the study population from Mysore, which included lineage 1 (*n* = 11), lineage 2 (*n* = 1), lineage 3 (*n* = 4), and lineage 4 (*n* = 10).

Further, the isolates were subjected to in silico spoligotyping and long-sequence polymorphism analysis (region of difference—RD) using SpolPred software of TB Profiler, and the results are as shown in Table 3. EAI (40%), followed by CAS (22%), were the main spoligotypes identified among both DM and non-DM patients.

To trace the evolutionary relationships between the clinical isolates, 48 best-quality genomes were subjected to pan-genome analysis and phylogenetic tree construction by the Roary pan-genome pipeline. Table 4 shows the core genome statistics used for tree construction, and Figure 3 shows the Roary matrix with distribution of core genes and accessory genes. Trees based on core and accessory genes were visualized with iToL, as shown in Figure 4 and Figure 5, respectively, revealing grouping of the isolates into specific clades, which matched with the lineage analysis by TB Profiler database. In addition, the phylogenetic trees did not reveal any striking genetic clusters of isolates within lineages 1, 2, 3, and 4.

### 3.3. M. tuberculosis Drug Resistance

Drug resistance of *M. tuberculosis* isolates by DM status is shown in Table 3. Drug resistance mutations were detected by PhyresSE and TB Profiler. Isoniazid resistance was seen in 5 of the 55 isolates (9%). Substitution of serine to threonine at codon 315 of the *katG* gene accounted for mutations in two DM and three non-DM patients. Five isolates (one DM and four non-DM) showed resistance to streptomycin (9%). Three of them accounted for mutations in *gidB*, with two of them showing substitution of valine to glycine at codon 65 and one having glycine-to-alanine substitution at codon 34. Two other isolates had *rrs* gene mutations. Of the 55 isolates, only 1 showed ethambutol resistance (2%), which had a mutation in the *embB* gene with substitution of glycine to aspartine at codon 406. Two isolates (one DM and one non-DM) had resistance to ethionamide (4%) with mutation of *ethA* gene with substitution of glycine to aspartine at codon 413.

## 4. Discussion

India has a high burden of TB, accounting for nearly a quarter of the global burden of TB. The prevalence is also high, estimated at 188/100,000 in 2021 [20]. The increasing prevalence of DM will exacerbate the incidence of TB, despite the efforts of the National Tuberculosis Elimination Program (NTEP) launched in 2020. DM occurring in an individual with LTBI may induce the latent infection to reactivate, or a new infection occurring someone with DM may lead to rapid progression to active disease. These outcomes are attributed to diminished protective immunity in those with DM. Here, we hypothesized that diminished immunity in DM subjects may enhance their susceptibility to a greater diversity of *M. tuberculosis* strains than in subjects without DM. We performed a detailed analysis of the WGS of *M. tuberculosis* strains isolated from patients with TB with or without DM. We found no significant difference in the M. tuberculosis spoligotypes isolated from DM vs. non-DM patients. Those with DM were all infected with the same spoligotypes found in non-DM TB patients. There was no difference in distribution of these genotypes with respect to geographical place of residence of the patients. Phylogenetic analysis revealed that the distribution of the genotypes from DM and non-DM TB patients clustered together. There was also no significant difference in drug resistance profiles in terms of DM status. 

The only previous WGS study from South India identified *M. tuberculosis* lineages 1 (Indo-Oceanic lineage) and 3 (East-African Indian lineage) as the predominant lineages, which occur at a substantially lower frequency elsewhere. The Indo-Oceanic lineage was the most common (70%), followed by the East-African Indian lineage (16%) [21]. Lineages 2 (East Asian lineage) and 4 (Euro-American lineage) are most common in Europe, Africa, and many other parts of the world [22]. Within India, lineage 3 is predominantly found in the North, while lineage 1 is common in the South [23]. We observed that the most common lineage detected among both the DM and non-DM subjects was the Indo-Oceanic lineage, which accounted for 40% of the isolates, while the Euro-American lineage accounted for 32% of the isolates. 

Although not significant, CAS was more common among TB patients with DM in our study. The Beijing strain, which is associated with greater risk for drug resistance [24,25], was seen in only one isolate (1.8%) in our study. The prevalence of the Beijing strain increases in states geographically situated northwards, with the highest prevalence observed in the state of Sikkim at 62.4% [26]. 

Our finding is similar to a study conducted in Kenya [27] which concluded that DM did not significantly increase *M. tuberculosis* genotype clustering among TB patients. A meta-analysis of six cohort studies of *M. tuberculosis* genotypes in those with DM concluded that clustering of DM in TB transmission chains has to be further investigated, due to several limitations pertaining to study setting and factors impacting cluster frequency [28].

The major limitation of this study is the small sample size of the TB patients and their isolates. The TB patients were identified over a period of 12 months. Furthermore, the analysis of the lineages and drug resistance was confined to in silico analyses of the WGS, which may have misclassified some of the isolates [29] However, other genotypic or phenotypic analyses of the *M. tuberculosis* strain are not likely to have yielded different results. A more granular analysis of the WGS to identify genetic features of the pan-genomes of the isolates from DM vs. non-DM TB patients may reveal features associated with DM or non-DM patients. Table 5 shows studies determining lineages and spoligotypes distributed across the world and specifically in India [30,31,32,33,34,35,36,37,38,39,40,41,42,43,44,45,46,47,48,49,50,51,52,53,54,55,56,57,58,59,60,61,62]. An overview of various studies conducted in different regions around the world to determine the predominant lineages and spoligotypes of *Mycobacterium tuberculosis*, offers valuable insights into the distribution and diversity of the strains in different populations, and also raises several points for critical discussion. Firstly, there is a predominance of the Euro-American lineage in several regions, such as Uganda, Botswana, Tanzania, and Argentina. The high prevalence of this lineage suggests a common ancestry and widespread dissemination, possibly through historical migration or colonization. However, without a comprehensive analysis of genetic variations within the Euro-American lineage, it is challenging to ascertain whether these strains originated from a single source or if multiple introductions occurred. Secondly, the predominance of specific spoligotypes within lineages raises questions about their epidemiological significance. For instance, the LAM spoligotype is prevalent in Botswana, Peru, and Kenya, while the T spoligotype is predominant in Botswana and Argentina. Understanding the transmission dynamics and potential virulence of these specific spoligotypes requires additional investigations, such as molecular typing techniques and population-based studies. Furthermore, the use of different methods for strain typing, including SNP typing, spoligotyping, MIRU-VNTR, WGS, and LSP typing, across the studies raises concerns about comparability and standardization. While each method has its advantages and limitations, it is crucial to establish consistent methodologies to ensure accurate comparisons between studies. Harmonizing the techniques used would facilitate a more comprehensive and standardized understanding of TB strain distribution worldwide. Another noteworthy observation is the significant regional variation within a country. For instance, in India, different spoligotypes dominate in various regions, such as Beijing type in Sikkim, EAI in Agra, and CAS in Delhi. These regional differences may be attributed to local transmission dynamics, genetic factors, or socioeconomic variations. Understanding the reasons behind this intranational heterogeneity would aid in developing targeted interventions and policies to control the spread of TB within specific regions. Additionally, the table highlights the wide use of spoligotyping as a typing method in several studies. While spoligotyping provides valuable information on the genetic diversity of TB strains, its limitations in discriminating closely related strains or accurately predicting transmission chains are well-documented. Therefore, integrating complementary techniques, such as MIRU-VNTR and WGS, can enhance the resolution and accuracy of strain typing, leading to a more comprehensive understanding of TB epidemiology. This summary of important studies provides a snapshot of TB strain distribution in different regions worldwide and highlights the predominance of specific lineages and spoligotypes in various populations, raising important questions about their origins, transmission dynamics, and potential implications for disease control. However, the variability in methods used across studies and the need for standardized approaches warrant further attention. Future research should aim to address these limitations and explore the underlying factors contributing to the observed strain diversity, ultimately aiding in the development of effective TB control strategies.

The study focused on understanding the impact of DM on the diversity of *M. tuberculosis* strains and drug resistance profiles. It was hypothesized that individuals with DM may be more susceptible to a greater diversity of *M. tuberculosis* strains due to diminished protective immunity. However, the analysis revealed no significant difference in the spoligotypes or drug resistance profiles between DM and non-DM TB patients. The predominant lineages detected were the Indo-Oceanic lineage and the Euro-American lineage, which aligns with previous studies conducted in South India. The findings suggest that there are no specific lineages more common among diabetics compared to non-diabetics unlike studies from Lima, Peru [13] which observed that subjects with diabetes and pre-diabetes had significant differences in mycobacterial strains causing disease. Drug resistance analysis revealed that isoniazid resistance was observed in 9% of the isolates, with specific mutations detected in the *katG* gene. Streptomycin resistance was found in 9% of the isolates, associated with mutations in the *gidB* and *rrs* genes. Ethambutol resistance was observed in only 2% of the isolates, with a mutation in the *embB* gene, while 4% isolates showed resistance to ethionamide with mutations in the *ethA* gene. Fortunately, there were no cases with rifampicin resistance. These findings provide valuable insights into the clinical characteristics, genomic diversity, and drug resistance profiles of *M. tuberculosis* isolates in the study population, contributing to our understanding of tuberculosis and its management.

However, the study’s limitations, including the small sample size and reliance on in silico analyses, should be considered. Further research with a larger sample size and more comprehensive genetic analyses is required. In future, we plan to explore the potential associations between DM and *M. tuberculosis* strains inspired by studies that used computational models [63] and comparative transcriptomics [64] designed to study the mechanisms of such relationships. This would provide a holistic view on this subject.

## 5. Conclusions

There was no difference in the genotypes or drug resistance profiles of *M. tuberculosis* isolated from DM vs. non-DM TB patients. The Indo-Oceanic lineage, followed by the Euro-American lineage were both similarly represented in DM and non-DM patients. Spoligotyping analysis observed that EAI and CAS were the most common spoligotypes. The analysis of the WGS based on current *M. tuberculosis* lineage classification schemes may not be sufficiently sensitive. However, the hypothesis that strains and lineages of *M. tuberculosis* among diabetics versus non-diabetics may differ cannot be rejected based on these preliminary study results. Further, to confirm the strength of these associations and to obtain a holistic view, we need comprehensive analysis, including comparative transcriptomics and computational analysis, to be performed on a larger population.

## Figures and Tables

**Figure 1 microorganisms-11-01881-f001:**
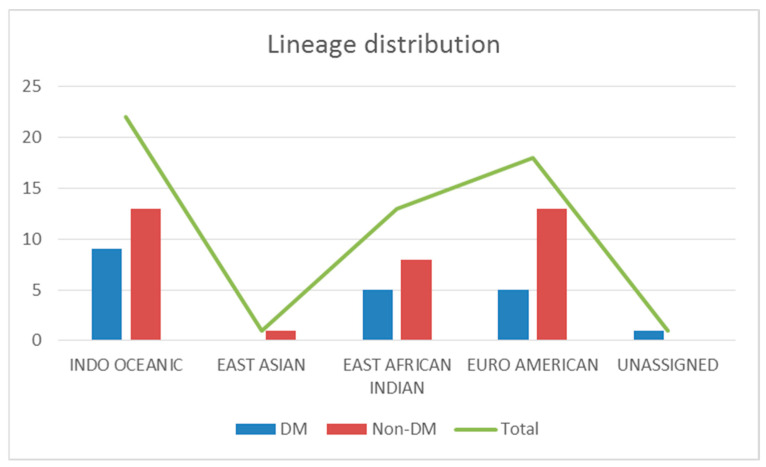
Distribution of lineages in the study population. Major lineage identified was Indo-Oceanic (*n* = 22) followed by Euro-American lineage (*n* = 18), East-African Indian lineage (*n* = 13), and East Asian lineage (*n* = 1). One particular isolate out of the fifty-five was identified as contaminated.

**Figure 2 microorganisms-11-01881-f002:**
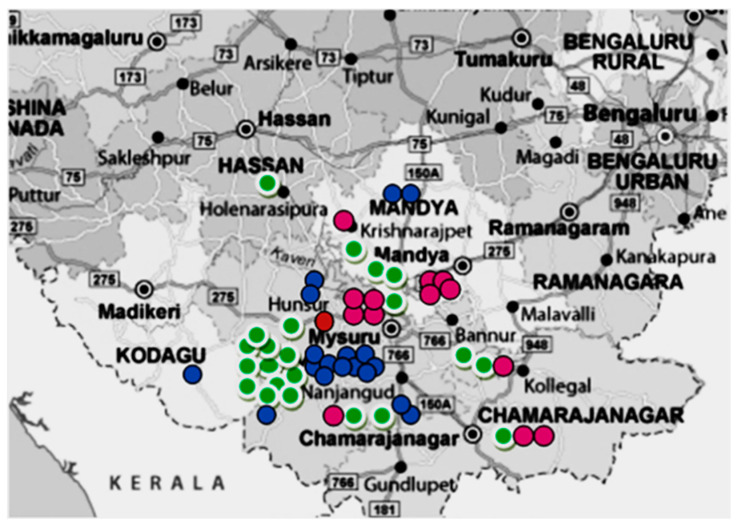
A pictorial representation of lineages in different taluks and districts of South Karnataka. (Lineage 1- Green; Lineage 2- Red; Lineage 3- Pink; Lineage 4- Blue). All 4 lineages were detected in the study population from Mysore, i.e., lineage 1 (*n* = 11), lineage 2 (*n* = 1), lineage 3 (*n* = 4), and lineage 4 (*n* = 10).

**Figure 3 microorganisms-11-01881-f003:**
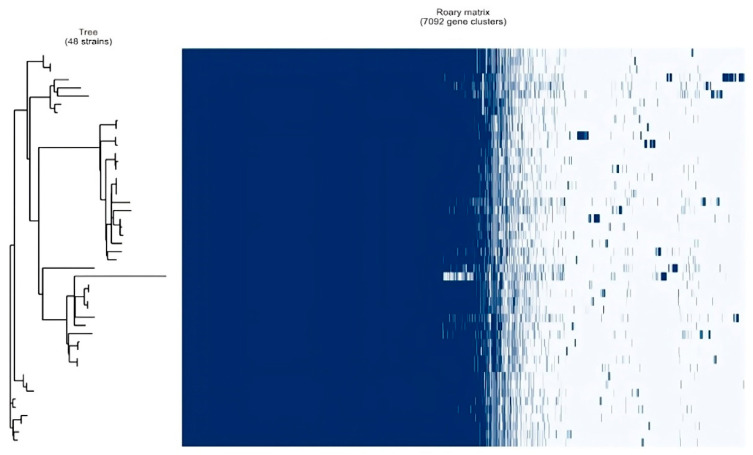
Roary matrix showing distribution of core genes and accessory genes.

**Figure 4 microorganisms-11-01881-f004:**
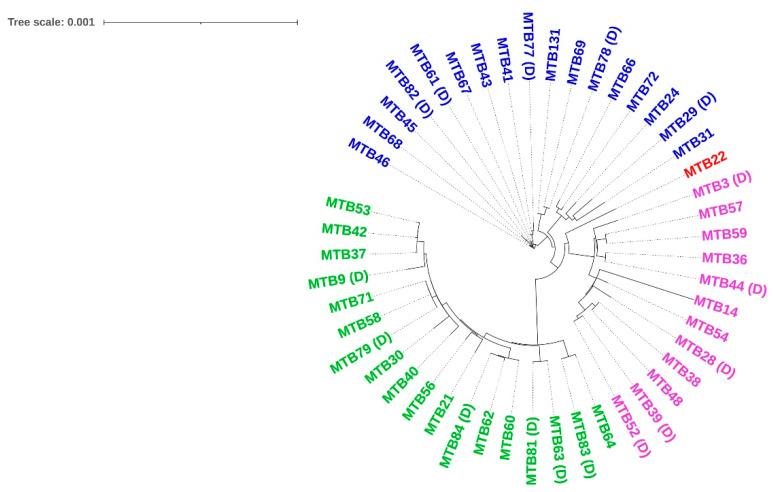
Phylogenetic tree constructed based on core genes. Isolates marked as (D) indicate TB-DM.

**Figure 5 microorganisms-11-01881-f005:**
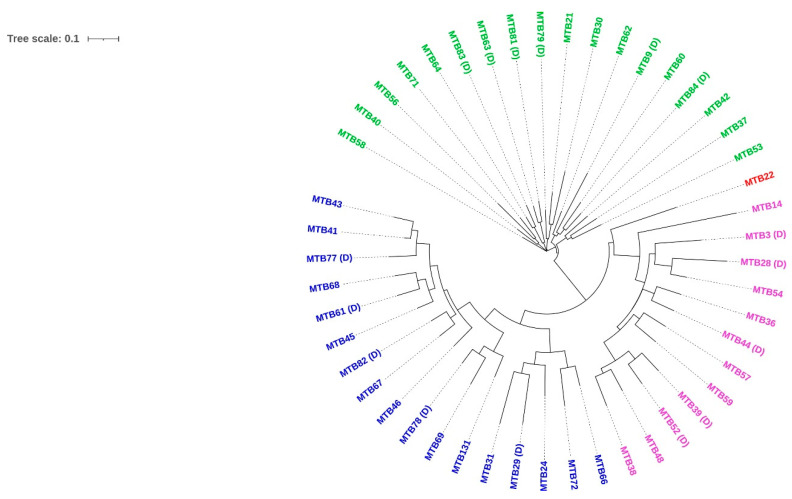
Phylogenetic tree constructed based on accessory genes. Isolates marked as (D) indicate TB-DM.

**Table 1 microorganisms-11-01881-t001:** Socio-demographic characters of study population (S—Significant, NS—not significant).

Patient Characteristics	Total	DM	Non-DM	*p*-Value
**Gender**				
Male	40	13 (23.63%)	27 (49.09%)	0.33 (NS)
Female	15	7 (12.72%)	8 (14.54%)	
**Age (years)**				
<30	10	0	10 (18.18%)	0.014 (S)
30–49	24	7 (12.72%)	17 (30.90%)	
50–69	16	11 (20%)	5 (9.09%)	
70+	5	2 (3.63%)	3 (5.45%)	
**Education**				
Primary school	12	3 (5.45%)	9 (16.36%)	0.64 (NS)
Middle school	2	0	2 (3.63%)	
Secondary school	10	5 (9.09%)	5 (9.09%)	
Pre-University	5	2 (3.63%)	3 (5.45%)	
Undergraduate	2	2 (3.63%)	0	
No qualification	24	8 (14.54%)	16 (29.09%)	
**Occupation**				
Student	3	0	3 (5.45%)	0.87 (NS)
Agriculturist	11	6 (10.90%)	5 (9.09%)	
Daily-wages laborer	6	3 (5.45%)	3 (5.45%)	
Business	9	3 (5.45%)	6 (10.90%)	
Housewife	5	2 (3.63%	3 (5.45%)	
Unknown/unemployed	21	6 (10.90%)	15 (27.27%)	
**Co-morbidities apart from DM**				
Hypertension	2	1 (5.26%)	1 (5.26%)	0.31 (NS)
Asthma	7	1 (5.26%)	6 (31.57%)	0.91 (NS)
Chronic obstructive pulmonary disease	1	1 (5.26%)	0	1 (NS)
Ischemic heart disease	2	1(5.26%)	1 (5.26%)	0.69 (NS)
Smoker	7	4 (21.05%)	3 (15.78%)	0.22 (NS)

**Table 2 microorganisms-11-01881-t002:** Distribution of lineages with respect to geographical locations of patients and their diabetic status (DM—diabetic, ND—non-diabetic). Lineage 1 (Indo-Oceanic) was the most common among both diabetic (*n* = 9) and non-diabetic populations (*n* = 13).

Lineage	Charmrajnagar		Hassan		Hd Kote		Hunsur		Kodagu		Kr Pete		Mandya		Mysore		Nanjangud		Srirangapatana		T.narsipura		Total
	ND	DM	ND	DM	ND	DM	ND	DM	ND	DM	ND	DM	ND	DM	ND	DM	ND	DM	ND	DM	ND	DM	
Euro American					1		2		1				1	1	6	4	2						18
East Asian															1								1
East African Indian	1	1									1		2	2	2	2	1				1		13
Indo Oceanic		1	1			1								3	8	3	2			1	2		22
	1	2	1	0	1	1	2	0	1	0	1	0	3	6	17	9	5	0	0	1	3	0	
Total	3		1		2		2		1		1		9		26		5		1		3		

**Table 3 microorganisms-11-01881-t003:** Comparison of spoligotypes/RD and drug resistance. NS—not significant.

Main Spoligotype	Region of Difference	Total	DM	Non-DM	*p*-Value
CAS	RD 750	12	6	6	NS
EAI	RD 239	22	8	14	0.31 (NS)
Beijing	RD 181	1	0	1	NS
T	RD 182	5	1	4	0.16 (NS)
T	RD 219	4	1	3	0.30 (NS)
LAM	RD 219	10	3	7	
Unassigned	-	1	1	-	0.16 (NS)
Drug resistance					
Not detected		46	17	29	0.83 (NS)
Isoniazid		3	1	2	1 (NS)
Streptomycin		2	0	2	NS
Isoniazid + Ethambutol		1	1	0	NS
Isoniazid + Streptomycin		1	0	1	NS
Ethionamide + Streptomycin		2	1	1	1 (NS)
MDR		0	0	0	NS
**Total**		55	20	35	

**Table 4 microorganisms-11-01881-t004:** Core genome statistics used for construction of phylogenetic tree.

Genes		Total Number
Core genes	(99% <= strains <= 100%)	3283
Soft core genes	(95% <= strains < 99%)	435
Shell genes	(15% <= strains < 95%)	575
Cloud genes	(0% <= strains < 15%)	2799
Total genes	(0% <= strains <= 100%)	7092

**Table 5 microorganisms-11-01881-t005:** Studies determining lineages and spoligotypes distributed across the world and specifically in India; SNP—Single-nucleotide polymorphisms, RT-PCR—Reverse-transcriptase polymerase chain reaction, LSP—Long-sequence polymorphism, RFLP—Restricted-fragment-length polymorphism, MIRU-VNTR—Mycobacterial interspersed repetitive unit variable number of tandem repeats.

An Overview of Various Studies Across the World:					
Study Conducted in	Year	Predominant Lineage	Predominant Spoligotype	Method	Reference in Discussion
Uganda, East Africa	2021	Euro-American (74.2%)	-	SNP typing by PCR	30
Botswana, South Africa	2019	Euro-American (81.9%)	LAM (33%), T (16%)	Spoligotyping and MIRU-VNTR	31
Tanzania, East Africa	2019	Euro-American (42.5%)	-	SNP typing by PCR	32
Ethiopia, East Africa	2021	Euro-American (61.6%)	-	LSP typing by PCR	33
Argentina, S.America	2018	Euro-American (99%)	T (35.9%), LAM (33.2%)	Spoligotyping	34
Peru, South America	2021	Euro-American (91.2%)	LAM (26.47%), Harleem (23.5%)	WGS	35
Colombia, S.America	2021	Euro-American (100%)	-	WGS	36
China	2017	East Asian (42.1%)	-	LSP typing and MIRU-VNTR	37
China	2021	East Asian (74.38%)	-	WGS	38
Shanghai	2022	East Asian (97.4%)	-	WGS	39
Japan	2021	East Asian (78.3%)	-	WGS	40
Vietnam	2019	East Asian (57.2%)	-	WGS	41
Thailand	2019	East Asian (44.6%), Indo-Oceanic (40%)	-	WGS	42
Philippines	2019	Indo-Oceanic (80.3%)	-	WGS	43
Malaysia	2021	Indo-Oceanic (93.8%)	-	WGS	44
South India (NIT, Delhi)	2017	Indo Oceanic (70%)	-	WGS	45
North India (JALMA, UP)	2021	East-African Indian (66.25%)	CAS (65%), Beijing (14.1%)	Spoligotyping	46
Study conducted in	Year	Predominant Spoligotype		Method	Reference
CMC Vellore, South India	2017	North Indian isolates—Beijing (23.4%), South Indian isolates—(EAI 43%)		Spoligotyping	47
JSSMC, Mysore, South India	2022	EAI (46%)		Spoligotyping	48
Sikkim, North India	2021	Beijing type (62.41%)		Spoligotyping and MIRU-VNTR	26
Varanasi, North India	2020	Beijing type (19.95%)		Spoligotyping	49
JALMA, Agra, North India	2019	EAI (51%), CAS (19%)		Spoligotyping and MIRU-VNTR	50
Madhya Pradesh, Central India	2019	CAS		Spoligotyping and RFLP	51
Bhopal, MP, Central India	2016	CAS (70%), EAI (30%)		Spoligotyping	52
AIIMS, Delhi, North India	2015	CAS (35.4%), EAI (24.2%)		Spoligotyping	53
Pondicherry, South India	2015	EAI (41.8%)		Spoligotyping	54
Delhi University, North India	2011	CAS		Spoligotyping and RFLP	55
Hinduja Hospital, Mumbai	2005	Beijing type (35%)		RFLP	56
Baba Atomic research, Mumbai	2005	CAS (30%), EAI (17%)		Spoligotyping	57
AIIMS, Delhi, North India	2012	CAS (57.27%)		Spoligotyping	58
Andhra Pradesh, South India	2011	CAS (40%), EAI (38%)		Spoligotyping	59
Bangladesh	2022	Beijing type (38%)		Spoligotyping	60
Mexico	2021	H (32%), T (23%)		Spoligotyping and MIRU-VNTR	61
Kenya	2017	CAS (28.9%), LAM & Beijing (17.6%)		Spoligotyping and MIRU-VNTR	62

## Data Availability

All data generated or analyzed during this study are included in this published article and are available from the corresponding author upon reasonable request.

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
