# Peer review of "Whole-Genome Sequencing of Mycobacterium tuberculosis Isolates from Diabetic and Non-Diabetic Patients with Pulmonary Tuberculosis"

_microorganisms, 2023, doi:10.3390/microorganisms11081881_

Round 1
Reviewer 1 Report
It is quite obvious that the authors at the initial stage of planning the study incorrectly set the task. There are practically no examples showing the effect of diabetes mellitus on the tuberculosis genotype. This is due to a simple fact. The primary infection by tuberculosis bacilli occurs before the age of 15, while diabetes mellitus appears on average 20 (or more) years later. From this point of view, it was much more correct for the authors to focus on the impact of DM on the development of MDR-TB. However, patients were selected for the study who had not received treatment prior to the start of the study. It is likely that existing data should be supplemented with samples from patients with MDR/XDR. The impact of DM in this case can be assessed by the rate of acquisition of drug resistance by TB strains of both category of patients.
Author Response
Thank you for the valuable suggestions. We have tried our best to address the concerns raised. Please find the response attached.

Reviewer 2 Report
This study is interesting and the manuscript is also well-written. As the authors states, the limited case number and old data in 2019 may limit its generalizability. In addition, I have several comments.
1. Pleaes add the use of systemic corticosteroid and inhaled corticosteroid in the demographic table.
2. Pleaes add more discussion about table 5.
3. Introduction was too long, so the description about LTBI may be deleted.
4. Pleaes add more discusison about the clinical implication of this study,
Author Response

(The authors gave the same response as above.)

Reviewer 3 Report
The submitted manuscript represent a good research idea, even though the manuscript is badly written. I suspect authors try to rush its writing and formatting. However, I will not reject it due to this, but authors really need to address the following topis for the manuscript to be considered:
- Italics - a lot of italics are missing in species name (even in the manuscript title!!!!)
- Abstract is too short - no mentioned or methods or results
- Even though authors are english native speakers, I think that the manuscript needs a careful revision, mainly in terms of grammar. For example, almost no commas are used!
- I think that the required format for references is [1] and not (1)
A lot of references are missing. Most relevant examples:
- line 22 - it is the first sentence of the manuscript, and a reference is needed
- methods - articles using similar methods should be cited, for simplification and clearance
Methods:
- Study design: what do you mean by "patients suspected to have pulmonary TB"?
- Sample processing: "The extracted DNA samples were then subjected to WGS." This sentence is irrelevant and non-informative.
- line 79 - "2nm in 10 ul". Please uniformize units (with or without space after numbers)
- a section explaining statistical analysis is needed - chi-square test, etc.
Results:
- Figures need to be uniformized in the manuscript. Figure 1 is a printscreen from excel, Figure 2 is a non-formatted printscreen of a table, Figure 3 lacks quality, and Figures 4 and 5 are badly cut images with part of the margins showing! This lack of quality and rigor is unacceptable for a journal of this prestige.
Discussion:
- A table is inserted in the middle of the discussion!!! Results should be moved to the Results section.
- Line 231: authors can not conclude simply that there was no differences! I suggest to include other kind of analysis (or at least suggest them as future work), as for example the ones included in the following citation:
* 10.1002/yea.3016
* 10.1093/femsyr/fox057
The mentioned works, even though applied to different organisms, include statistical and computational analysis that would benefit the kind of results shown in this manuscript. I recommend authors to give it a try, to enrich their conclusions, or at least cite them as future work for a follow up study. Without this, I think the current manuscript does not include enough novelty!
English language, especially in terms of formatting, are very bad!! Authors just rush the writing of the manuscript, without any rigor!
Author Response

(The authors gave the same response as above.)

Round 2
Reviewer 2 Report
The authors response well, so I have no more comment.
Author Response
Thank you for your valuable suggestions and guidance which has significantly improved the manuscript quality.
Reviewer 3 Report
The manuscript improved a lot with the introduction of the required changes.
As a final change, authors need to move the table legends to before the table, and not after.
Author Response
As per your suggestion the table legends have been moved to the top.
Thank you for your guidance in preparation of a better version of our manuscript.